# Management of Glycemia during Acute Aerobic and Resistance Training in Patients with Diabetes Type 1: A Croatian Pilot Study

**DOI:** 10.3390/ijerph20064966

**Published:** 2023-03-11

**Authors:** Marul Ivandic, Maja Cigrovski Berkovic, Klara Ormanac, Dea Sabo, Tea Omanovic Kolaric, Lucija Kuna, Vjera Mihaljevic, Silvija Canecki Varzic, Martina Smolic, Ines Bilic-Curcic

**Affiliations:** 1Department of Internal Medicine, University Hospital Osijek, 31000 Osijek, Croatia; 2Faculty of Kinesiology, University of Zagreb, 10000 Zagreb, Croatia; 3Department of Pharmacology, Faculty of Medicine Osijek, J. J. Strossmayer University of Osijek, 31000 Osijek, Croatia; 4Department of Pharmacology and Biochemistry, Faculty of Dental Medicine and Health Osijek, J. J. Strossmayer University of Osijek, 31000 Osijek, Croatia

**Keywords:** type 1 diabetes mellitus, aerobic training, resistance training, flash glucose monitoring, hypoglycemia

## Abstract

(1) Background: The increased risk of developing hypoglycemia and worsening of glycemic stability during exercise is a major cause of concern for patients with type 1 diabetes mellitus (T1DM). (2) Aim: This pilot study aimed to assess glycemic stability and hypoglycemic episodes during and after aerobic versus resistance exercises using a flash glucose monitoring system in patients with T1DM. (3) Participants and Methods: We conducted a randomized crossover prospective study including 14 adult patients with T1DM. Patients were randomized according to the type of exercise (aerobic vs. resistance) with a recovery period of three days between a change of groups. Glucose stability and hypoglycemic episodes were evaluated during and 24 h after the exercise. Growth hormone (GH), cortisol, and lactate levels were determined at rest, 0, 30, and 60 min post-exercise period. (4) Results: The median age of patients was 53 years, with a median HbA1c of 7.1% and a duration of diabetes of 30 years. During both training sessions, there was a drop in glucose levels immediately after the exercise (0′), followed by an increase at 30′ and 60′, although the difference was not statistically significant. However, glucose levels significantly decreased from 60′ to 24 h in the post-exercise period (*p* = 0.001) for both types of exercise. Glycemic stability was comparable prior to and after exercise for both training sessions. No differences in the number of hypoglycemic episodes, duration of hypoglycemia, and average glucose level in 24 h post-exercise period were observed between groups. Time to hypoglycemia onset was prolonged after the resistance as opposed to aerobic training (13 vs. 8 h, *p* = NS). There were no nocturnal hypoglycemic episodes (between 0 and 6 a.m.) after the resistance compared to aerobic exercise (4 vs. 0, *p* = NS). GH and cortisol responses were similar between the two sessions, while lactate levels were significantly more increased after resistance training. (5) Conclusion: Both exercise regimes induced similar blood glucose responses during and immediately following acute exercise.

## 1. Introduction

Patients with T1DM and uncontrolled glucose levels can develop cardiovascular complications, while glucose-lowering treatment strategies can prevent or postpone their appearance. In those patients, exercise is advised for reducing the risk of several chronic conditions [1,2]. Nevertheless, there are some associated risks with exercise, such as fluctuating blood glucose levels from hyperglycemia to hypoglycemia [3]. In the past couple of decades, much attention has been placed on the relationship between T1DM and exercise, which led to new ideas for the treatment of T1DM [4].

Although exercise is advocated by the American Diabetes Association (ADA) as a means of treatment for people with diabetes, around 63% of individuals with T1DM are inactive [5]. The main reason is the fear of potential adverse events that may occur during exercise, the most important being hypoglycemia. However, insulin dosage can be reduced and glycemic stability improved, minimizing the risk of hypoglycemia [5,6,7].

Additionally, aerobic exercise improves the quantity and function of insulin receptors on skeletal muscles and fat and increases the overall number of cellular glucose transporters, which leads to increased glucose uptake and a corresponding faster decrease of blood glucose during the activity [8,9]. In healthy individuals, during aerobic exercise, insulin release decreases while glucagon secretion increases, resulting in relatively stable glycemia. However, in T1DM, due to a lack of beta cell function, the only insulin available is the exogenous one. Its concentrations can be higher than endogenously produced insulin, and therefore, blood glucose levels are largely dependent on timing, quantity, and type of insulin administered. Thus, drops in blood glucose levels occur frequently and rapidly. Moreover, the liver produces less glucose in T1DM individuals who exercise because of higher blood insulin concentrations. Therefore, ingesting carbohydrates prior to aerobic exercise as well as reducing/withholding insulin doses might be a necessary strategy to avoid hypoglycemia [10,11,12]. On the other hand, exaggerated reduction or even skipping of insulin dosage, as well as consuming excessive amounts of carbohydrates, causes hyperglycemia before and during the exercise, possibly leading to ketosis [13,14,15].

During resistance training, insulin and glucagon cease to be the key energy regulation source, leading to increased lactate production due to non-oxygen-dependent glucose decomposition (anaerobic metabolism). Catecholamine levels increase in response to anaerobic changes push glucose production and release from the liver, causing blood glucose levels to rise and thus minimizing the risk of blood glucose drop during exercise. Nevertheless, hypoglycemia may develop for an extended period after exercise [12,16,17].

Better maintenance of glucose stability is associated with resistance exercise compared to aerobic exercise, although it can induce rebound hyperglycemia, probably due to increased secretion of counter-regulatory hormones [17,18]. Conversely, short and vigorous anaerobic exercise (for example, short sprints) or high-intensity interval training (HIIT) decreases the decline in blood glucose levels and thus has protective effects [19]. Minimization of hypoglycemia risk is attributed only to the exercise session itself; however, the current data for post-exercise hypoglycemia after anaerobic exercise do not support a clear conclusion [15,18,20,21].

Defining the optimal type and time of exercise is extremely difficult, even in healthy individuals. Patients with T1DM, besides individual preferences for different exercise types, need to consider the additional factor during physical activity, which is the maintenance of target glucose levels, although utilizing technology tools such as intermittent or continuous glucose monitoring, insulin pumps, and closed loop systems significantly improved the management of T1DM. Therefore, a better understanding of the impact of aerobic exercise, resistance training, or HIIT on glycemic variability, hypoglycemia, hyperglycemia, and energy expenditure is important for creating optimal strategies for better management of diabetes and a positive impact on HbA1c.

The aim of this study was to assess glycemic levels and hormonal changes after acute aerobic versus resistance training using intermittent continuous glucose monitoring in patients with T1DM.

## 2. Materials and Methods

### 2.1. Participants

We conducted a randomized crossover study including 14 untrained adult patients with T1DM for at least one year, with HbA1c of less than 9% and a stable insulin regimen for the preceding 3 months. Patients with frequent hypoglycemia and those with hypoglycemia unawareness, patients on corticosteroid therapy, and those with chronic renal impairment (eGFR < 90 mL/min), active diabetic retinopathy, foot ulcers, and liver disease were excluded. The Ethical Committee of the Faculty of Medicine, University of Osijek, and the Ethical Committee of the Clinical Hospital Center Osijek approved the study protocol (R2-4787/2019 and 16 April 2019 respectively), and the investigation was conducted according to the Declaration of Helsinki. All patients gave their written informed consent before inclusion. Patients were randomized according to the type of exercise (aerobic and resistance exercise) with a recovery period of three days between group changes. Data on physical activity, diabetes glycemic stability, and hypoglycemia history were assessed before inclusion. All participants had an intermittently scanned continuous glucose monitoring (isCGM) FreeStyle Libre sensor placed 90 days prior to the study entry and were instructed to perform capillary blood glucose tests when measuring hypoglycemia with isCGM and to measure ketones in cases of blood glucose of >15 mmol/L. The study protocol was performed 48 h post sensor change to avoid any errors due to lower accuracy in the first 24 h or at the end of their useful lifespan. Interstitial glucose levels were monitored during a 24 h post-exercise period. Participants withheld from exercise for 24 h before or during the washout period. The study design scheme is presented in Figure 1. IsCGM data were handled according to recommendations from the international consensus on time in range by Battelino et al. [17].

### 2.2. Experimental Design

Participants arrived at the clinical research facility between 17:00 and 18:00 h. The following protocol for maintaining normoglycemia during exercise was applied: 75% dose reduction of fast-acting insulin administered prior to consuming a meal containing 1 g of carbohydrate/kg 60 min before physical activity; post-workout meal containing 1 g of the low glycemic index (GI) carbohydrate/kg 60 min after physical activity with 50% dose reduction of fast-acting insulin; bedtime snack containing 0.3 g of carbohydrate/kg with low GI and omission of prandial insulin [8]. An exact meal plan based on patients’ weight was created by a certified nutritionist. After a standardized 5 min warm-up of the main muscle groups, participants undertook aerobic/resistance exercises. The threshold for hypoglycemia was ≤3.9 and for hyperglycemia, >10.0 mmol/L. A hypoglycemic episode registered with isCGM was confirmed with a standard blood glucose meter. Data on insulin doses, food intake, and levels of physical activity for 24 h prior to exercise were obtained, and insulin doses and food intake were copied between experimental sessions of different types of exercise.

Blood plasma was sampled at rest and 0 (immediately post exercise), 30, and 60 min (recovery phase) after cessation of the exercise, and blood glucose, lactate, GH, and cortisol levels were determined. Plasma was analyzed immediately after sampling. Measurements of blood glucose, HbA1c, and lactates were performed by routine assays using an automatic analyzer Olympus AV 640 (Olympus, Beckman Coulter, Inc., Brea, CA, USA). GH was determined using Beckman Coulter Access Dxi Chemiluminescence Immunoassay (CLIA), and cortisol was determined using the Alinity Chemiluminescence Microparticle Immunoassay system (CMIA), Abbott Diagnostics.

### 2.3. Exercise Protocol

All participants performed two workout sessions with a 72 h recovery period in between and no additional activity.

The aerobic exercise session lasted 45 min and consisted of walking on a treadmill to achieve a heart rate of 70% intensity as determined by the estimated HRmax. After a 5 min warm-up at a heart rate of 50 to 60% of HRmax (calculated for males as (202 − (0.55 × age) × 0.50) to (202 − (0.55 × age) × 0.60) and females as (216 − (1.09 × age)) × 0.50) to (216 − (1.09 × age)) × 0.60), followed a workout session for 30 min at a heart rate of 70–75% of HRmax (calculated for males as (202 − (0.55 × age) × 0.70) to (202 − (0.55 × age) × 0.75) and females as (216 − (1.09 × age)) × 0.70) to (216 − (1.09 × age)) × 0.75). The last 10 min were the same as the warm-up. Resistance training was conducted for 45 min using bodyweight and bar exercises including 10 exercises for different muscle groups in 3 sets of 12, 12, and 10 repetitions with a 30 s break between sets and a 2 min break between exercises. Resistance exercises included floor chest presses, overhead presses, bar squats, standing bent-over rows, and lying triceps extensions. The perceived rate of exertion was 5 to 6 on a scale of 1 to 10, which was below the expected anaerobic threshold and in a zone where, if needed, conversation could still be carried with effort. The anaerobic exercise target zone was considered to begin when 80% of the maximal heart rate was reached, which is considered an anaerobic threshold. The exercise protocol is shown in Figure 2.

### 2.4. Statistical Methods

Categorical data were presented in absolute and relative frequencies. Differences in categorical variables were tested by the McNemar–Bowker test or the marginal homogeneity test. The normality of the distribution of continuous variables was tested by Shapiro–Wilkinson test. Numerical data are described by median and interquartile range. Differences in numerical variables were tested by the Wilcoxon test. The association score is given by the Spearman correlation coefficient. All *p*-values are two-sided. The significance level was set to alpha = 0.05. MedCalc Statistical Software version 19.1.7 (MedCalc Software Ltd., Ostend, Belgium; https://www.medcalc.org; accessed on 4 March 2020) was used for statistical analysis.

## 3. Results

Fourteen participants performed exercises. The median age of patients was 53 years, with a median HbA1c of 7.1%, duration of diabetes of 30 years, and a BMI of 24.5 kg/m^2^ (Table 1).

Differences in glucose levels at different time points during aerobic and resistance exercise are summarized in Table 2. During both training sessions, there was a significant drop in blood glucose levels when comparing the pre-exercise period with 12 h and 24 h time points (*p* = 0.001). There was a drop in glucose levels immediately after the exercise (0′), followed by an increase at 30′ and 60′, although the difference was not statistically significant.

Glycemic levels were comparable prior and after exercise for both training sessions. There were no differences between the two exercise types (Table 3).

No differences in the number of hypoglycemic episodes, duration of hypoglycemia, nor average glucose level in 24 h post-exercise period were observed when comparing aerobic vs. resistance exercise. Time to hypoglycemia onset after resistance exercise compared to aerobic did not reach statistical significance (Table 4).

There were 14 hypoglycemic episodes after aerobic and resistance exercise (*p* > 0.99) and 4 nocturnal hypoglycemic episodes after aerobic training, but none after resistance training (*p* = 0.13) (Table 5).

Plasma cortisol and GH responses were similar at four different time points following resistance and aerobic exercise session. Lactate levels were significantly higher after resistance compared to the aerobic training session at 0, 30, and 60 min (*p* < 0,05); data obtained from blood plasma (Table 6).

There was no change in lactate levels during aerobic training; however, a significant increase was noted immediately after cessation of resistance training (0 min) with complete normalization after 60 min (Figure 3). Cortisol levels were at their lowest 60 min after training (123 nmol/L; *p* = 0.02) in relation to all three-time points in the aerobic session, both before and immediately after aerobic exercise. Following resistance training, a decrease in cortisol was observed, whereas the peak level was registered at rest (270 nmol/L) with a tendency to drop (230–198–185 nmol/L, *p* = 0.01) over 60 min (Figure 4). An increase in growth hormone levels was observed after both aerobic and resistance training, followed by a decrease during the recovery phase (both *p* < 0.001); data obtained from blood plasma (Figure 5).

## 4. Discussion

Lately, overcoming the problem of hypoglycemia and maintaining normoglycemia during exercise in T1D seems quite attainable, not only due to recently available technological tools such as continuous/flash glucose monitoring systems, insulin pumps, and closed loop systems, but also specifically developed protocols regarding insulin dose adjustments and carbohydrate intake in order to prevent hypoglycemia [19,22]. Still, it is not clear whether one type of exercise is more suitable than another [23]. Resistance exercise causes a smaller drop in glucose levels during the activity itself, yet larger reductions of glycemic levels in the post-exercise period were observed compared to aerobic training [22,24]. On the other hand, in some studies, resistance exercise was also associated with a modest increase in glycemia [18], while intense anaerobic workouts led to an increase in glucose levels [20].

Our results demonstrated similar advantages of resistance and aerobic training regarding rebound hyperglycemia and maintenance of glycemic stability over a 24 h post-exercise period. This could be explained by the study design implementing a protocol including low-GI meals 60 min prior to and after each type of exercise with the administration of reduced boluses by 25% and 50% of calculated bolus, respectively. In a previously published study investigating the impact of low- versus high-GI food on post-treadmill exercise glucose levels, the low-GI meal better prevented post-exercise hyperglycemia compared to the high-GI meal. However, both the high- and low-GI meals protected all patients from early hypoglycemia, but the risk of nocturnal hypoglycemia remained [11]. In addition, Campbell et al. showed that rapid-acting insulin reduction of 25% pre-treadmill exercise and 50% post-treadmill exercise maintained glucose levels and protected against early- (<8 h) but not late-onset hypoglycemia [25]. In addition, our study showed an increased occurrence of post-exercise hypoglycemic episodes; however, duration, and the number of episodes were similar regardless of the type of exercise, confirming the results of a recently published trial comparing aerobic and resistance training using CGM [22]. However, the time to hypoglycemia onset was prolonged for at least 8 h, confirming the usefulness of the applied protocol for hypoglycemia avoidance.

Furthermore, all our participants were treated with multiple daily injections (MDI) using ultra-long-acting insulin, with no adjustments of basal insulin doses as recommended in previous studies [26,27]. Reduction in basal insulin dose might promote hyperglycemia at multiple points during the day (especially for patients using long- or ultra-long-acting insulin). Because it still reduces hypoglycemia risk during and after exercise, it could be particularly recommended for patients performing prolonged and intense activity, which was not the case in our study [26,27].

Several studies demonstrated that performing aerobic exercise in the afternoon or evening increases the risk of hypoglycemic episodes, especially nocturnal ones [26,28,29,30]. In our study, time to hypoglycemia onset was prolonged after the resistance as opposed to aerobic training (13 vs. 8 h), explaining the lack of nocturnal hypoglycemia after the resistance compared to aerobic exercise (4 vs. 0). This difference was not statistically significant but could point to the potential benefits of resistance versus aerobic exercise, especially if it is performed in the afternoon. Our results are further substantiated by the recent meta-analysis focusing on the delayed effects of different exercise modalities. Additionally, Valli et al. demonstrated that there is a reduced risk of hypoglycemia if exercise is performed in the morning rather than in the afternoon with a 50% rapid-acting insulin reduction, but no definite benefits of resistance exercise were determined [30,31]. Based on the current recommendations, exercise should ideally be performed in the morning or at noon. However, many employed patients prefer exercising in the afternoon or evening; therefore, in this subset of patients, resistance training could be preferable to aerobic exercise [28].

IsCGM has been proven to be clinically valuable, reducing the risks of hypoglycemia, hyperglycemia, and glycemic variability (GV) and improving patient quality of life for a wide range of patient populations and clinical indications [32,33,34,35,36]. Still, isCGM also comes with a set of drawbacks. There is 10–15 min of lag time, because glucose is measured in interstitial fluid, as shown in previous studies [26,37]. If the glucose values are changing rapidly (e.g., intensive exercise, eating, etc.) the lag time is longer. In our participants, the maintenance of relatively good glycemic stability after both types of training could be partially attributed to the isCGM usage or close glucose monitoring, including trend arrows through which patients could easily follow glucose fluctuations over time, allowing timely detection and appropriate action if inadequate glucose values were observed [26,37,38].

As was expected, the GH response to both types of exercise were similar, with a significant increase immediately after the cessation, followed by a decrease in the recovery phase. The increase of GH levels facilitated by all types of exercise is well-documented [39,40,41]. GH response to stimuli depends on several factors, such as duration and intensity of physical activity, training, and fitness state [41]. It seems that in our study, the intensity of both types of exercise was comparable, as well as the training state of participants, leading to similar GH response. This is further substantiated by findings in several studies showing that during typical resistance training (RT) programs, such as the one used in this study, adults spend the majority of time at moderate intensity, regardless of BMI or age [42].

In our study, a decrease in cortisol concentrations during both types of training was observed, whereas previous studies reported the opposite effect, with higher cortisol levels during training [18,22]. Variations in circadian rhythm could justify a lack of response or even a decline of cortisol levels to both types of exercise even when they are performed in the afternoon [43]. Therefore, the lack of difference between glycemic stability in both exercise types could be explained by similar responses of counter-regulatory hormone in both exercise modes given that meals, basal insulin, and short-acting insulin reduction were standardized.

Lactate levels were significantly increased following resistance training, while they remained completely unaffected after the aerobic session. This finding confirms a significant anaerobic component to RE reported in other studies [44,45]. Apparently, a rise in catecholamine secretion during RE induces muscle glycogenolysis enhancing glycogen decomposition mediated through phosphorylase α, leading to increased lactate production [44,46]. High lactate levels slow down glycogen utilization in the muscles, which could, in turn, help prevent early post-exercise hypoglycemia in patients with T1DM, as was the case in our study.

The limitation of the current study is the small sample size. On the other hand, randomization to the cross-over design and detailed study protocol controlling both insulin doses and carbohydrate intake (low GI), add to the study’s strength. Moreover, measuring cortisol, GH, and lactate levels helped in the interpretation of glycemia levels after the aerobic and anaerobic exercise, suggesting that the higher lactate levels seen after resistance training potentially attenuate the decline in blood glucose levels, especially in the early post-exercise period, via stimulation of gluconeogenesis.

## 5. Conclusions

Both resistance and aerobic exercise induced similar blood glucose responses during and immediately following acute exercise. Resistance training seems to be a more favorable choice in patients exercising in the afternoon or evening considering the prolonged time to hypoglycemia onset; however, this finding needs to be further investigated using a larger sample size. In addition, low GI meals and bolus dose reduction in combination with isCGM usage successfully prevented early post-exercise hyperglycemia in both types of exercise.

## Figures and Tables

**Figure 1 ijerph-20-04966-f001:**
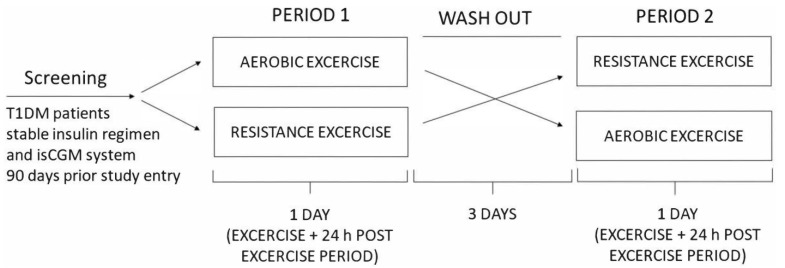
Study design scheme.

**Figure 2 ijerph-20-04966-f002:**
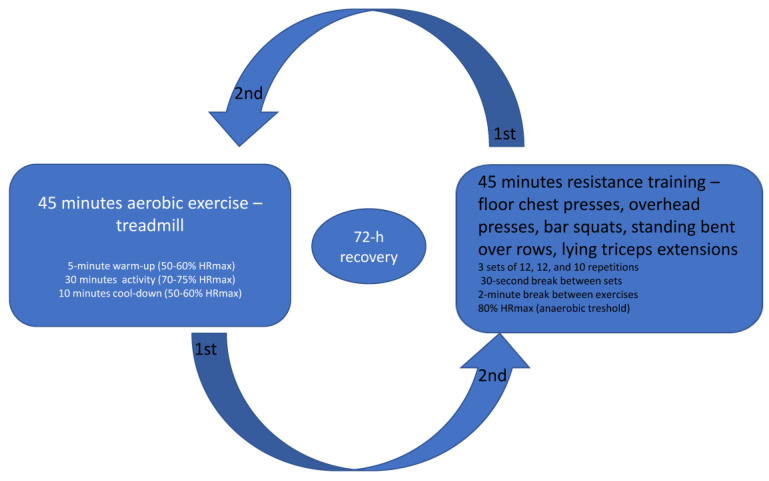
Aerobic and resistance training protocols were separated by 72 h recovery between the workout sessions.

**Figure 3 ijerph-20-04966-f003:**
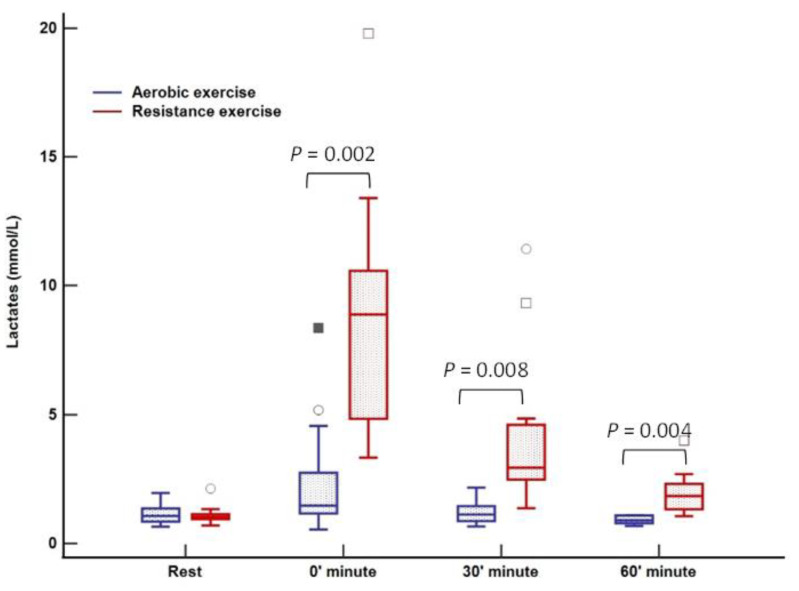
Lactate response to aerobic versus resistance training. The box-and-whisker plot displays the following: the median value (solid line within the box); the interquartile range (2 ends of the box); either the 5th or 95th percentile values or the smallest and largest observations (2 lines with caps extending from the box, the so-called whiskers); and any extreme outliers.

**Figure 4 ijerph-20-04966-f004:**
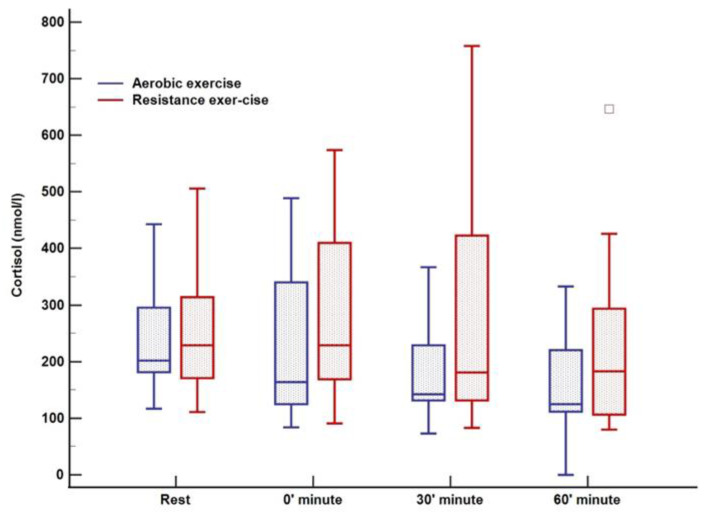
Cortisol response to aerobic versus resistance training. The box-and-whisker plot displays the following: the median value (solid line within the box); the interquartile range (2 ends of the box); either the 5th or 95th percentile values or the smallest and largest observations (2 lines with caps extending from the box, the so-called whiskers); and any extreme outliers.

**Figure 5 ijerph-20-04966-f005:**
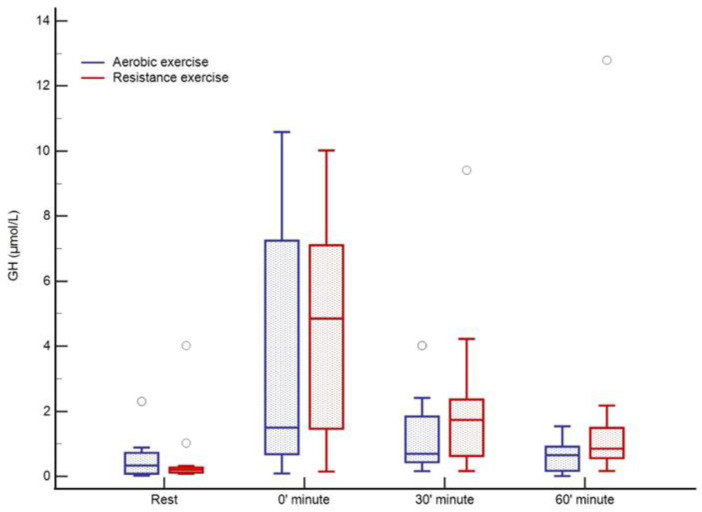
GH response to aerobic versus resistance training. The circles represent the extreme outliers.

**Table 1 ijerph-20-04966-t001:** Baseline characteristics of participants.

Parameter	Median (IQR) ^1^	Minimum–Maximum
Age (years)	53 (43–57)	22–60
Duration of diabetes (years)	30 (20–40)	1–53
HbA1c %	7.1 (6.35–7.83)	5.4–9
BMI (kg/m^2^)	24.5 (23.07–26.42)	21.2–29.7
Gender (N)	Females 9	Males 5

^1^ IQR: interquartile range.

**Table 2 ijerph-20-04966-t002:** Differences in glucose levels at different time points during aerobic and resistance training sessions; data from IsCGM.

Aerobic Exercise	Glucose Level Median (IQR)	*p* *	Resistance Exercise	Glucose Level Median (IQR)
Rest	9.2 (8.2–11.2)	**0.001** ^†^	Rest	10.3 (6.4–11.8)
0′ min	7.9 (6.4–11.1)		0′	8.1 (7.3–11.4)
30′ min	8.5(7.3–12.7)		30′ minute	8.75 (7.6–12.4)
60′ min	9.6 (8.6–12.9)		60′ minute	9.45 (7.2–12.8)
6 h	7.7 (6.1–11.0)		6 h	7.3 (6.2–9.9)
12 h	7.4 (4.9–10.6)		12 h	7.7 (6–13)
24 h	6.9 (5.4–8.2)		24 h	6.4 (4.4–9.2)

* Friedman test (post hoc Conover); ^†^ *p* < 0.05: rest vs. 12 h; rest vs. 24 h. IQR: interquartile range.

**Table 3 ijerph-20-04966-t003:** Differences in glucose levels at different time points comparing aerobic and resistance exercise, measured by IsCGM.

	Glucose Level Median (IQR)	Hodges–Lehmann Median Difference	95% CI	*p* *
Prior AE	9.2 (8.2–11.2)	−0.15	−2.5 to 2.2	0.95
Prior RE	10.3 (6.4–11.8)			
0′ AE	7.9 (6.4–11.1)	1.10	−1.1 to 2.9	0.36
0′ RE	8.1 (7.3–11.4)			
30′ AE	8.5 (7.3–12.7)	−0.15	−2.25 to 2.05	0.90
30′ RE	8.75 (7.6–12.4)			
60′ AE	9.6 (8.6–12.9)	−0.70	−3.25 to 1.7	0.54
60′ RE	9.45 (7.2–12.8)			
6 h AE	7.7 (6.1–11.0)	−0.25	−3.2 to 1.7	0.79
6 h RE	7.3 (6.2–9.9)			
12 h AE	7.4 (4.9–10.6)	1.1	−1.4 to 3.4	0.33
12 h RE	7.7 (6–13)			
24 h AE	6.9 (5.4–8.2)	0.75	−1.7 to 2.5	0.54
24 h RE	6.4 (4.4–9.2)			

IQR: interquartile range; * Wilcoxon test; AE: aerobic exercise; RE: resistance exercise.

**Table 4 ijerph-20-04966-t004:** Differences in the number and duration of hypoglycemic episodes and average glucose level 24 h after training, depending on the type of exercise; data extracted from IsCGM.

	Median (IQR)	Hodges–Lehmann Median Difference	95% CI	*p* *
Number of hypoglycemia 24 h after aerobic exercise	1 (0–2)	−0.5	−1 to 0.5	0.25
Number of hypoglycemia 24 h after resistance exercise	1 (0–1)			
Time to hypoglycemia onset after aerobic exercise (hours)	13 (5–20)	4	−3 to 9.5	0.30
Time to hypoglycemia onset after resistance exercise (hours)	18 (13.5–21.5)			
Duration of hypoglycemia (min)–24 h after aerobic exercise	60 (0–220)	−17	−105 to 30	0.50
Duration of hypoglycemia (min)–24 h after resistance exercise	65 (0–120)			
Average glucose level 24 h after aerobic exercise (mmol/L)	7.8 (6.78–9.75)	−0.15	−1 to 0.7	0.57
Average glucose level 24 h after resistance exercise (mmol/L)	7.3 (6.55–9.03)			

IQR: interquartile range; * Wilcoxon test.

**Table 5 ijerph-20-04966-t005:** Differences in the number of total and nocturnal hypoglycemic episodes 24 h after training, depending on the type of exercise.

Number of patients	Number (%)		Diff.	95% CI	*p* *
	Aerobic Exercise	Resistance Exercise			
Total hypoglycemic episodes	14	14	7.1%	−29.7 to 43.9	>0.99
Nocturnal hypoglycemic episodes	4	0	30.8%	5.7 to 55.9	0.13

* McNemar–Bowker test.

**Table 6 ijerph-20-04966-t006:** Lactate, cortisol, and GH response to aerobic versus resistance training.

Time Point	Median (IQR)		Difference ^†^	95% CI	*p* *
	Aerobic	Resistance			
**Lactates (mmol/L)**					
Rest	1.12 (0.88–1.48)	1.02 (0.91–1.17)	−0.10	−0.51 to 0.24	0.73
0 min	1.47 (1.20–3.35)	9.01 (4.62–10.67)	5.79	2.48 to 9.42	**0.002**
30 min	1.12 (0.83–1.44)	2.73 (2.35–5.60)	2.37	0.80 to 5.89	**0.008**
60 min	0.93 (0.83–1.09)	1.83 (1.21–2.05)	0.74	0.41 to 1.25	**0.004**
**Cortisol (nmol/L)**					
Rest	202 (181.5–295.5)	270 (201.3 to 337.3)	13.5	−49 to 158.5	0.58
0 min	164 (125–340.3)	230 (178.8–415.5)	36.5	−59 to 188.5	0.41
30 min	142.5 (131.5–229)	198 (140–466)	52.5	−18.5 to 274	0.15
60 min	123 (110.5–237)	185 (128.5–315.5)	46.5	−6 to 127.9	0.08
**GH (μmol/L)**					
Rest	0.343 (0.103–0.727)	0.203 (0.126–0.304)	−0.04	−0.681 to 0.541	0.76
0 min	1.46 (0.672–7.437)	4.85 (1.465–7.105)	1.26	−0.289 to 3.085	0.17
30 min	0.612 (0.409–1.844)	1.798 (0.625–3.399)	0.48	−0.197 to 5.583	0.18
60 min	0.723 (0.161–0.974)	0.853 (0.560–1.489)	0.31	−0.490 to 0.959	0.30

IQR: interquartile range; * Wilcoxon test; ^†^ Hodges–Lehmann median; GH: growth hormone.

## Data Availability

The datasets used and analyzed in the present study can be made available from the author (ibcurcici@mefos.hr) upon reasonable request.

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
