# Peer review of "Management of Glycemia during Acute Aerobic and Resistance Training in Patients with Diabetes Type 1: A Croatian Pilot Study"

_ijerph, 2023, doi:10.3390/ijerph20064966_

Round 1

Reviewer 1 Report

This study aimed to assess glycemic stability and hypoglycemic episodes during after aerobic versus resistance exercises using a flash glucose monitoring system in patients  with T1DM.

There are several defects in this study:

1) The number of patients is very limited.

2) The data are not well presented.

Author Response

Thank you for your effort to review the Manuscript. We have incorporated your comments.

Reviewer 2 Report

This document is interesting, and the information it provides is essential. However, it could be improved with schemes chosen by the participant's selection and schemes where the differences between resistance exercise versus aerobics are visualized.

Author Response

(The authors gave the same response as above.)

Reviewer 3 Report

General comments

I read with interest, a report entitled “Management of glycemic control during acute aerobic and resistance training in patients with diabetes type 1 - a Croatian pilot study” Indeed, the study is of interest provided the difficult in managing glucose levels in patients with type 1 diabetes. The study aims to enhance the management of type 1 diabetes. I just have a few comments:

Specific comments

The major limitation is the low number of patients (n=14). Which explains the use “a pilot study” Perhaps also mention this limitation within the abstract as it is very significant?

Within the abstract, revisit the sentence “However, glucose levels significantly decreased from 60’ to 24h time points (P=0,001) for both types of exercise” Try to make this statement clear to the reader.

Introduction, should clearly highlight the significance of looking at both types of exercise.

I don’t understand why the title reads “Management of glycemic control…” rather choose one between “Management of glycemia…” or “… glycemic control…”

Check if abbreviations are written at first time and described accurately throughout the manuscript.

Author Response

(The authors gave the same response as above.)

Round 2

Reviewer 1 Report

I did not see any improvement in the data presentation. All the data are presented in tables. There should be figures to clearly show the data.

Author Response

Dear Reviewer please find the changes made according to your suggestions.
